# Secondary Organic Aerosol Formation from the Oxidation of Decamethylcyclopentasiloxane at Atmospherically Relevant OH Concentrations

Sophia M. Charan[1], Yuanlong Huang[1], Reina S. Buenconsejo[1], Qi Li[2], David R. Cocker III[2], and John H. Seinfeld[1]

[1]California Institute of Technology, Pasadena, California 91125, United States
[2]University of California – Riverside, Riverside, California 92521, United States

**Correspondence:** seinfeld@caltech.edu

**Abstract.** Decamethylcyclopentasiloxane (D5, $C_{10}H_{30}O_5Si_5$) is measured at ppt levels outdoors and ppb levels indoors. Primarily used in personal care products, its outdoor concentration is correlated to population density. Since understanding the aerosol formation potential of volatile chemical products is critical to understanding particulate matter in urban areas, the secondary organic aerosol yield of D5 was studied under a wide range of OH concentrations and, correspondingly, OH exposures using both batch-mode chamber and continuously run flow tube experiments. These results were comprehensively analyzed and compared to two other secondary organic aerosol (SOA) yield datasets from literature. It was found that the SOA yield from the oxidation of D5 is extremely dependent on either the OH concentration or exposure. For OH concentrations of $\lesssim 10^7$ molec $cm^{-3}$ or OH exposures of $\lesssim 2 \times 10^{11}$ molec s $cm^{-3}$, the SOA yield is largely <5% and usually ~1%. This is significantly lower than SOA yields previously reported. Using a two-product absorptive partitioning model for the upper-bound SOA yields, the stoichiometric mass fraction and absorptive partitioning coefficients are, for the first product, $\alpha_1 = 0.056$ and $K_{OM,1} = 0.022$ $m^3$ $\mu g^{-1}$; for the second product, they are $\alpha_2 = 7.7$ and $K_{OM,2} = 4.3 \times 10^{-5}$ $m^3$ $\mu g^{-1}$. Generally, there are high SOA yields (> 90%) at OH mixing ratios of $5 \times 10^9$ molec $cm^{-3}$ or OH exposures above $10^{12}$ molec s $cm^{-3}$.

## 1 Introduction

Present in outdoor mixing ratios as high as ~40 ppt, decamethylcyclopentasiloxane (D5, $C_{10}H_{30}O_5Si_5$) has been observed in cities, rural areas, and the Arctic (Buser et al., 2013, 2014; Ahrens et al., 2014; McLachlan et al., 2010; Xu et al., 2019). D5 is used in personal care products, as well as for industrial purposes (Mackay et al., 2015); in 2004, over 17000 tons were used in the then European Union (Safron et al., 2015). Outdoor observations of D5 are population-dependent (Janechek et al., 2017; Gkatzelis et al., 2021) and this dependence is sufficiently reliable that it can be used to tease out personal care product emission

patterns from other common urban emissions (Coggon et al., 2018). The impact of D5 does not stop at population centers; its long atmospheric lifetime means that it is even found in areas with low population densities.

Likely more than 90% of the D5 used is emitted into the atmosphere (Balducci et al., 2012; Hughes et al., 2012), though much of this may be first emitted indoors and only later exchanged to the outdoors: in an engineering classroom in the U.S. in 2014, ~30% by mass of the total volatile organic compounds (VOCs) were D5 (Tang et al., 2015). In an athletic center in the
25 morning, D5 mixing ratios exceeded 6 ppb and emissions were attributed to the humans in the room (Finewax et al., 2020). Even the international space station contains trace amounts of D5 in the air (Carter et al., 2015).

Given the abundance of D5 in the ambient atmosphere, it is important to understand its fate. The major loss source of D5 is reaction with the hydroxyl radical; losses by reaction with $NO_3$ and $O_3$ are negligible (Atkinson, 1991; Alton and Browne, 2020). Global losses by reaction with Cl are less than 5%, though can be higher in polluted areas (Alton and Browne, 2020).
The half-life of D5 outdoors is between 3.5 and 7 days, depending on the assumed global average OH concentration and the exact method used to determine the reaction rate of OH with D5 (Safron et al., 2015; Xiao et al., 2015; Alton and Browne, 2020). Outside, both wet and dry deposition of D5 are negligible and methylsiloxanes do not photolyze in the actinic region (Janechek et al., 2017; Hobson et al., 1997).

Chandramouli and Kamens (2001) detected a single product from OH oxidation with D5, 1-hydroxynonamethylcyclopentasiloxane.
A study of 1-hydroxynonamethylcyclopentasiloxane partitioning found a large temperature, aerosol seed type, and humidity dependence (Latimer et al., 1998). Though, an investigation by Wu and Johnston (2017) found both dimer products and ring-opened species from the OH oxidation of D5. SOA from all D5 oxidation products was investigated by Janechek et al. (2019).

Previous studies investigating the secondary organic aerosol (SOA) yields of D5 oxidation with OH found values between 8 and 50% (Janechek et al., 2019; Wu and Johnston, 2017). SOA yields are the ratios of the mass of organic aerosol formed
to the mass of the precursor reacted. This is a wide range, and the conditions for these experiments were performed at OH concentrations much higher than those in the ambient atmosphere. By measuring the SOA formation potential of D5, we can better understand the contribution of volatile chemical products (VCPs) to aerosol levels in urban areas.

VCPs are a major (and perhaps majority) source of secondary organic aerosol in some U.S. cities (McDonald et al., 2018; Gkatzelis et al., 2021). Resolving uncertainties in the mass of SOA formed from VCPs is critical for refining SOA estimates
and for creating policy to reduce SOA non-compliance in urban areas (Burkholder et al., 2017).

For many years, researchers have used flow reactors, usually run at steady-state, and atmospheric chambers, usually run in batch mode, to understand SOA formation and the SOA yields of various compounds (Bruns et al., 2015). Batch mode is where all reactants are added before oxidation and, during each experiment, the evolution of the reactor's contents are tracked in time. While many results agree between the two methods of analysis, different reactors have varying benefits and operating
conditions (e.g., OH concentrations, experiment length, precursor concentrations, humidity values). One must account for the particular attributes of the different reactors when extrapolating to the atmosphere.

The chamber experiments conducted in this study were performed for multiple hours at OH concentrations representative of what is found in the atmosphere. Since these experiments are time-limited, and D5's outdoor half-life is multiple days, the

chamber experiments tend to have OH exposures slightly less than what is representative of ambient conditions. OH exposure
is the quantity of OH concentration over time and is a measure of atmospheric aging (Renbaum and Smith, 2011).

To understand SOA yields at higher OH concentrations, a flow reactor was run at steady-state. The residence time of this reactor was short (on the order of minutes), but the OH mixing ratios were higher than used in the chamber experiments.

It is well established that OH concentration and exposure are not necessarily interchangeable (Renbaum and Smith, 2011; Liu et al., 2011; McNeill et al., 2008). Additionally, there is precedent for studying an overlapping range of OH exposure using
both environmental chambers and flow reactors. For example, Lambe et al. (2015) showed that oxidation experiments over a range of OH exposures can be comparable between both types of reactors.

We start with a discussion of results from these two reactors, which show agreement when either the OH concentrations or exposures overlap. Then, we provide two-product absorptive partitioning parameters and fits for the data collected here. We close by comparing these results to other SOA yield studies in the literature: that of Wu and Johnston (2017) and Janechek
et al. (2019).

## 2   Methods

Chamber experiments (C1–8) were performed in a temperature-controlled 19 m$^3$ FEP Teflon Environmental Chamber run in batch mode. The uncertainty associated with the reported chamber temperature is $< 0.5°C$. The chamber is hung in an enclosure, to reduce charge on the surface of the chamber, and is surrounded by ultraviolet lights centered at ~350 nm. Since the walls
of the chamber are not rigid and data were collected continuously, the chamber decreased slightly in volume throughout the experiment, but never by more than 15%.

Prior to each chamber experiment, the contents of the chamber were flushed with air stripped of ozone, nitrogen oxides, water vapor, and organic carbon for $> 24$ h. Hydrogen peroxide ($H_2O_2$), when used as an OH source (C1–7), was injected by flowing air at 5 Lpm over liquid $H_2O_2$ in a ~42°C water bath to obtain an $[H_2O_2] \approx 2$ ppm. For experiment C8, an evacuated
glass bulb was filled with methyl nitrite ($CH_3ONO$) and diluted with nitrogen. The bulb was then flushed into the chamber with nitrogen to obtain a mixing ratio of ~600 ppb in the chamber. $CH_3ONO$ forms OH as described in Schwantes et al. (2019).

D5 (99%, TCI America) was injected into the chamber for experiments C1–8 at room temperature by flowing nitrogen through a glass bulb at 5 Lpm for $> 60$ min. To obtain the desired initial surface area concentration, a sonicated, 0.06 M (0.15 M for C2) $(NH_4)_2SO_4$ solution was atomized to create aerosol that was then dried, passed through a TSI Model 3088 soft
x-ray neutralizer, and injected into the chamber. For C7, no aerosol was injected. For experiments C5–7, NO (506.9 ppm $\pm$ 2%, Airgas Specialty Gases, Certified Standard) was injected prior to the beginning of the experiment to achieve initial NO mixing ratios between 80 and 100 ppb. The estimated uncertainty of $[NO]_0$ is 5 ppb. During C5 and C6, 1 ppb min$^{-1}$ of NO was injected from the inception of radiation to the end of the experiment. All experiments began with $[NO_2]_0 = 0$ ppb.

Experiments at higher OH mixing ratios were conducted in the Caltech Photooxidation Flow Tube (CPOT, Huang et al.,
2017) at steady-state and a constant flow rate of 4.88 Lpm and 23.0±0.1°C. The mean residence time of the CPOT was $671 \pm 15$ s and the diffusivity was $15 \pm 2$ cm$^2$ s$^{-1}$, as calculated with a step injection of $SO_2$ using Equation 4 in Huang and

Seinfeld (2019). For experiments F9–15, clean air flowed through an ozone generator (UVP, 97-0067-01); for F16–19, $O_2$ flowed through the same generator to create higher concentrations of $O_3$. The 254 nm lights photolyze $O_3$ to form $O(^1D)$, which reacts with $H_2O$ to form 2OH. After conditions were changed in the CPOT, no results were collected for at least 2 h. Data were averaged over between 1 and 11 h. D5 was injected through a syringe pump (Harvard Apparatus).

For all experiments, the concentration of D5 was measured with an HP 6890N gas chromatograph with a flame ionization detector (GC-FID) and a DB-5 column. Prior to the beginning of oxidation for the chamber experiments, all contents of the reactor were left to sit for 4 h (2.8 h for C7) and the initial concentration of D5 was taken as the mean concentration during this time. For the CPOT experiments, the initial concentration of D5 was calculated by measuring the outlet flow with lights off, no water source, and the absence of $O_3$. For F9, the change in D5 was sufficiently small that it was within the uncertainty. For calculating the SOA yield for this experiment, we used the OH exposure calculated from the change in $SO_2$ concentration to find the change in D5 (7 ppb).

To calibrate the GC-FID, a small Teflon bag was filled with 35 ppm of D5 and later diluted to 9 ppm. This bag was sampled using the GC-FID, and the concentration was verified with a Fourier transform infrared absorption (FT-IR) spectrometer with a 19 cm path length and absorption cross sections from the Pacific Northwest National Laboratory (PNNL) database. To minimize vapor-wall-loss to the FT-IR enclosure, multiple samples were taken until a consistent spectrum was achieved.

Gas-phase oxidation products were evaluated with a $CF_3O^-$ chemical ionization mass spectrometer (CIMS) equipped with a Varian 1200 triple quadrupole mass analyzer. Concentrations of NO and $NO_2$ were measured with a Teledyne Nitrogen Oxide Analyzer (Model T200) and $O_3$ was found with a Horiba Ambient Monitor. Temperature and humidity were determined using a Vaisala HMM211 probe.

Aerosol volume was measured by a custom-built scanning mobility particle sizer (SMPS) with a 3081 TSI Differential Mobility Analyzer (DMA) and a TSI 3010 butanol condensation particle counter (CPC). The sheath flow rate was 2.64 Lpm and the aerosol flow rate from the chamber was 0.515 Lpm. A voltage scan from 15 to 9875 V was performed in 240 s every 330 s. Aerosol from the chamber flowed through an x-ray source to provide a known charge distribution, and the size distributions were determined using the data inversion method described by Mai et al. (2018). Experiment C2 required a logarithmic fit to the largest particles present, as described in Charan et al. (2020), which is the source of the higher SOA yield uncertainty than in the other experiments (see Table 1). Conversions to mass concentration were performed by assuming that the aerosol density was $1.52 \pm 0.04$ g cm$^{-3}$, which was the density calculated at $[OH] \approx 9.4 \times 10^9$ molec cm$^{-3}$ in a flow reactor (Xu and Collins, 2021) using an Aerosol Particle Mass Analyzer and a SMPS system as described in Malloy et al. (2009).

**Table 1.** Experimental conditions. For the CPOT experiments, the upper estimate of the wall-deposition-corrected SOA yield is shown in parentheses.

| Label | Reactor | $[NO]_0$ (ppb) | Contin. NO Injection? | OH source | $[OH]$ (molec cm$^{-3}$) | OH exposure (molec s cm$^{-3}$) | T (°C) | RH (%) | $[D5]_0$ (ppb) | $1 - \frac{[D5]_{end}}{[D5]_0}$ | $[\text{Surf Area}]_0$ ($10^3$ μm$^2$ cm$^{-3}$) | SOA Yield |
|---|---|---|---|---|---|---|---|---|---|---|---|---|
| C1 | Chamber | 0 | No | $H_2O_2$ | $4.5 \times 10^6$ | $9 \times 10^{10}$ | 26.6 | <5 | $497 \pm 5$ | 0.18 | $3.6 \pm 0.3$ | $1.5 \pm 1.5\%$ |
| C2 | Chamber | 0 | No | $H_2O_2$ | $3.8 \times 10^6$ | $8 \times 10^{10}$ | 26.5 | <5 | $298 \pm 3$ | 0.17 | $5.1 \pm 0.3$ | $5.7 \pm 8.0\%$ |
| C3 | Chamber | 0 | No | $H_2O_2$ | $2.2 \times 10^6$ | $6 \times 10^{10}$ | 27.6 | <5 | $30 \pm 1$ | 0.09 | $0.8 \pm 0.1$ | $0 \pm 0.3\%$ |
| C4 | Chamber | 0 | No | $H_2O_2$ | $1.6 \times 10^6$ | $3 \times 10^{10}$ | 17.7 | <5 | $580 \pm 5$ | 0.08 | $2.4 \pm 0.2$ | $2.6 \pm 4.0\%$ |
| C5 | Chamber | 82 | Yes | $H_2O_2$ | $5.0 \times 10^6$ | $1.3 \times 10^{11}$ | 26.6 | <5 | $696 \pm 9$ | 0.22 | $3.9 \pm 0.3$ | $0.7 \pm 0.7\%$ |
| C6 | Chamber | 86 | Yes | $H_2O_2$ | $4.3 \times 10^6$ | $9 \times 10^{10}$ | 26.6 | <5 | $650 \pm 6$ | 0.18 | $1.1 \pm 0.1$ | $0.2 \pm 0.3\%$ |
| C7 | Chamber | 76 | No | $H_2O_2$ | $5.5 \times 10^6$ | $1.3 \times 10^{11}$ | 23.6 | <5 | $591 \pm 2$ | 0.24 | 0 | $0 \pm 0.1\%$ |
| C8 | Chamber | 84 | No | $CH_3NO_2$ | $10^6$–$10^{8.3}$ | $2.3 \times 10^{11}$ | 26.6 | <5 | $603 \pm 5$ | 0.38 | $0.6 \pm 0.1$ | $0 \pm 0.1\%$ |
| F9 | CPOT | 0 | No | $O_3$ | $2.0 \times 10^7$ | $1.4 \times 10^{10}$ | 23.0 | 2 | $262 \pm 10$ | 0.03 | 0 | $1.1(1.9) \pm 1.1\%$ |
| F10 | CPOT | 0 | No | $O_3$ | $2.3 \times 10^8$ | $1.5 \times 10^{11}$ | 23.0 | 3 | $262 \pm 10$ | 0.26 | 0 | $1.8(2.9) \pm 0.2\%$ |
| F11 | CPOT | 0 | No | $O_3$ | $5.0 \times 10^8$ | $3.3 \times 10^{11}$ | 23.0 | 4 | $262 \pm 10$ | 0.28 | 0 | $6.0(9.2) \pm 0.6\%$ |
| F12 | CPOT | 0 | No | $O_3$ | $2.3 \times 10^8$ | $1.5 \times 10^{11}$ | 23.0 | 3 | $262 \pm 10$ | 0.38 | 0 | $4.6(6.7) \pm 0.4\%$ |
| F13 | CPOT | 0 | No | $O_3$ | $1.2 \times 10^9$ | $7.8 \times 10^{11}$ | 23.0 | 10 | $262 \pm 10$ | 0.60 | 0 | $14(19) \pm 1\%$ |
| F14 | CPOT | 0 | No | $O_3$ | $1.5 \times 10^9$ | $1.0 \times 10^{12}$ | 23.0 | 16 | $262 \pm 10$ | 0.67 | 0 | $24(32) \pm 2\%$ |
| F15 | CPOT | 0 | No | $O_3$ | $1.6 \times 10^9$ | $1.1 \times 10^{12}$ | 23.0 | 33 | $262 \pm 10$ | 0.71 | 0 | $35(49) \pm 2\%$ |
| F16 | CPOT | 0 | No | $O_3$ | $4.7 \times 10^9$ | $3.2 \times 10^{12}$ | 23.0 | 25 | $246 \pm 3$ | 0.97 | 0 | $109(157) \pm 7\%$ |
| F17 | CPOT | 0 | No | $O_3$ | $4.8 \times 10^9$ | $3.2 \times 10^{12}$ | 23.0 | 30 | $246 \pm 3$ | 0.98 | 0 | $110(158) \pm 7\%$ |
| F18 | CPOT | 0 | No | $O_3$ | $4.7 \times 10^9$ | $3.1 \times 10^{12}$ | 23.0 | 23 | $82 \pm 3$ | 1.00 | 0 | $102(138) \pm 6\%$ |
| F19 | CPOT | 0 | No | $O_3$ | $4.9 \times 10^9$ | $3.3 \times 10^{12}$ | 23.0 | 33 | $82 \pm 3$ | 1.00 | 0 | $94(128) \pm 6\%$ |

Uncertainty estimates for all the instruments used in this study were determined as described in Charan et al. (2020). For the chamber experiments, particle-wall-deposition corrections were performed by calculating a diameter-independent first-order exponential fit ($\beta = 1$–$7 \times 10^{-4}$ min$^{-1}$) to the particle volume concentration during the 3 h prior to the onset of oxidation and applying that correction to the rest of the experiment. This method was chosen because it aligns with a diameter-dependent fit as determined using the method in Charan et al. (2018) but is simpler and because, for the chamber experiments, minimal organic aerosol formed and, so, the particle diameters changed insignificantly throughout the duration of the experiment. For experiment C7, in which no initial aerosol was present, no aerosol was generated throughout the experiment and so no correction was necessary to determine an apparent SOA yield of 0.

For the CPOT experiments, an upper estimate of the wall-deposition-corrected SOA mass was calculated with the inverse of the particle-size-dependent penetration efficiency of the flow-tube component of the reactor (data from Fig. 9d in Huang et al. 2017). Since particles nucleated in the CPOT, the penetration efficiency of only the flow-tube component (and not the static mixer prior to the region of reaction) was used. The penetration efficiency, however, is based on the entire flow tube and nucleated particles may not form immediately at the beginning of the flow-tube component; thus, the wall-deposition correction performed here is an upper bound of the correction. Note that this correction also neglects particle growth throughout the reactor and any particle-particle coagulation. Diagrams of the CPOT and its static mixer are in Huang et al. (2017).

SOA yield (Y) is defined by $Y = \frac{\Delta SOA_{corr}}{\Delta D5}$, where $\Delta SOA_{corr}$ is the wall-deposition-corrected change in the aerosol mass concentration and $\Delta D5$ is the mass concentration of reacted D5. Calculations were performed as described by Charan et al. (2020) and with the assumption that a particle, once deposited on the reactor wall, no longer acts as a condensation sink (Trump et al., 2016). Note that since so little aerosol was formed during the chamber experiments, this assumption had a negligible effect on the chamber results. For the CPOT experiments, any deviation from this assumption would have prevented the reactor from reaching steady-state.

Table 1 gives the upper estimate of the wall-deposition-corrected SOA yield for the CPOT experiments. The lower bound of the SOA yields of the CPOT experiments is the reported uncertainty subtracted from the uncorrected Y and the upper bound is the sum of the corrected Y and the reported uncertainty.

While the vapor-wall-deposition lifetime of D5 to the chamber walls was estimated to be on the order of weeks, the propensity of vapor-wall-deposition of the oxidation products is not extensively investigated in this study. Even at high initial seed surface area concentrations, the SOA yield is still quite small (see Fig. S1). Alton and Browne (2020) estimated that, for their unseeded ~1 $m^3$ FEP Teflon chamber, 5% of the ester product of D5 oxidation might partition to the chamber walls during the reaction. Though other products may have higher wall-loss rates, if 5% of the oxidation products were lost to the chamber walls in C1–8, the SOA yields would still be within the reported uncertainty and smaller than expected previously. The CPOT reactor is operated at steady-state (see Fig. S2) and, therefore, any oxidation products that are sufficiently high volatility and in equilibrium with the bulk flow (i.e., not lost permanently to the quartz walls) do not need a vapor-wall-deposition correction. While very low-volatility compounds may be lost to the reactor's walls, as was seen by Krechmer et al. (2020) in a continuously run Teflon reactor, we do not expect there to be significant irreversible gas-phase wall loss of siloxanes or their oxidation products in this reactor. Note that Wu and Johnston (2017) did see higher SOA yields in seeded experiments in their steady-state PFA Teflon reactor than in unseeded ones, indicating that there may be some irreversible loss, even when run continuously.

For chamber experiments that employed $H_2O_2$, the OH concentration was calculated by fitting the gas-phase D5 concentration to a first-order exponential, fixing the initial point of the fit as the initial D5 concentration (fits had $R^2 > 0.75$), and using the value for the reaction rate of OH with D5, $k = 2.1 \pm 0.1 \times 10^{-12}$ $cm^3$ $molec^{-1}$ $s^{-1}$, which was measured using the relative rate method at $297 \pm 3$ K (Alton and Browne, 2020). Note that other experimental evaluations of the reaction rate of OH with D5 that use the relative rates method vary by less than a factor of 2 (the reasons for this difference are not known), which would not affect the order of magnitude of the OH concentration estimate (Kim and Xu, 2017; Safron et al., 2015; Xiao et al., 2015). OH is the major loss source in the atmosphere and, we expect, in these experiments; note that there was no Cl in these experiments (Atkinson, 1991; Alton and Browne, 2020). The ozone concentration did not affect the SOA yield results: C7 and F9, which were performed at substantially different $O_3$ concentrations, still gave similar results for the SOA yield (0±0.1% and 0.8±0.8% with an upper wall-deposition-corrected bound of 1.4%, respectively). For experiments performed with $NO_x$ (Appendix A), there was no observed $NO_x$ dependence.

For C8, in which $CH_3NO_2$ served as the OH source, the sharp decrease in the D5 mixing ratio immediately after the commencement of radiation, followed by a gradual decrease in its concentration, indicates that two OH concentrations were relevant for this experiment ([OH]=$2 \times 10^8$ molec $cm^{-3}$ at the beginning of the experiment and [OH]=$1 \times 10^6$ molec $cm^{-3}$ at

its end). Since the D5 concentration was measured every ~21 min, and the pulse with high OH concentrations occured within the first 30 min of oxidation, the initial OH concentration is estimated with a two-point first-order exponential fit to the initial concentration and the first data point (12.3 min into radiation). The second OH concentration is estimated with a first-order exponential fit of the second point (33.3 min into radiation) to the end of the experiment.

    OH exposure was calculated, for chamber experiments (C1–8) and experiments from Wu and Johnston (2017), as

$$170 \quad [\mathrm{OH}] * t = \frac{-1}{k_{\mathrm{OH+D5}}} \ln\left(\frac{[\mathrm{D5}]_{\mathrm{end}}}{[\mathrm{D5}]_0}\right), \tag{1}$$

where $k_{\mathrm{OH+D5}} = 2.1 \times 10^{-12}$ cm$^3$ molec$^{-1}$ s$^{-1}$ (Alton and Browne, 2020). For the CPOT experiments (F9–19), OH exposure was calculated as

$$[\mathrm{OH}] * t = \frac{-1}{k_{\mathrm{OH+SO_2}}} \ln\left(\frac{[\mathrm{SO_2}]_{\mathrm{end}}}{[\mathrm{SO_2}]_0}\right), \tag{2}$$

where $k_{\mathrm{OH+SO_2}} = 9 \times 10^{-13}$ cm$^3$ molec$^{-1}$ s$^{-1}$ for a CPOT setup with $SO_2$ instead of D5 and otherwise identical flows and
conditions, as in the method described by Janechek et al. (2019). Uncertainties in the OH exposure for the CPOT experiments are $5 \times 10^{10}$ molec s cm$^{-3}$. The correlation between [$H_2O$] and OH exposure used to find the OH exposure for experiments F9–19 is plotted in Fig. S3. Since the $O_3$ concentrations differed in F9–15 and F16–19, the correlation between the [$H_2O$] and the OH exposure is also different. Note that, for F9–19, the OH exposure calculated using Equation 2 is about twice that calculated using Equation 1. This effect may be because of a lower OH concentration in the D5 system due to the reaction of OH with the
later D5 oxidation products or the aerosol surfaces. So, for F9–19, the OH exposure is calculated with Equation 2. Experiments C1–8 and those from Wu and Johnston (2017) use Equation 1, but have a positive uncertainty equal to their calculated OH exposure. They have a negative uncertainty of $4 \times 10^{10}$ molec s cm$^{-3}$. This does not include the potential additional uncertainty in the value of $k_{OH+D5}$ as calculated using similar methods in different laboratories. Note that, for C1–8, OH concentration is calculated independently of the OH exposure. For F9–19, OH concentration is the ratio of the OH exposure to the residence
time of the reactor. The uncertainty in the OH concentrations for the chamber and CPOT experiments are $5 \times 10^5$ molec cm$^{-3}$ and $1 \times 10^6$ molec cm$^{-3}$, respectively.

    In order to determine the absorptive partitioning coefficients, C1-3, C5-8, and F9-17 were fit to a two-product model (Odum et al., 1996):

$$Y = M\left(\frac{\alpha_1 K_{OM,1}}{1 + M K_{OM,1}} + \frac{\alpha_2 K_{OM,2}}{1 + M K_{OM,2}}\right), \tag{3}$$

where $Y$ is the SOA yield; $M$ is the organic aerosol mass concentration; $\alpha_1$ and $\alpha_2$ are the stoichiometric mass fractions of products 1 and 2, respectively; and $K_{OM,1}$ and $K_{OM,2}$ are the absorptive partitioning coefficients for products 1 and 2, respectively. Fits were performed for starting points that varied from $10^{-4}$ to $10^4$ and from $10^{-10}$ m$^3$ μg$^{-1}$ to $10^{10}$ m$^3$ μg$^{-1}$ for the $\alpha$s and $K_{OM}$s, respectively. The fits with the highest R$^2$ were chosen.

## 3 Results

### 3.1 Agreement between the CPOT and chamber experiments

The SOA yields from the CPOT and chamber experiments vary from 0 to 110% (158% at the upper bound of the wall-deposition-corrected value), an even wider range than that reported by the literature of 8–50% (Table 1 shows all experimental conditions and SOA yields). However, the measured SOA yields for these experiments (and those in the literature) correlate with the OH concentrations and the OH exposures (Fig. 1).

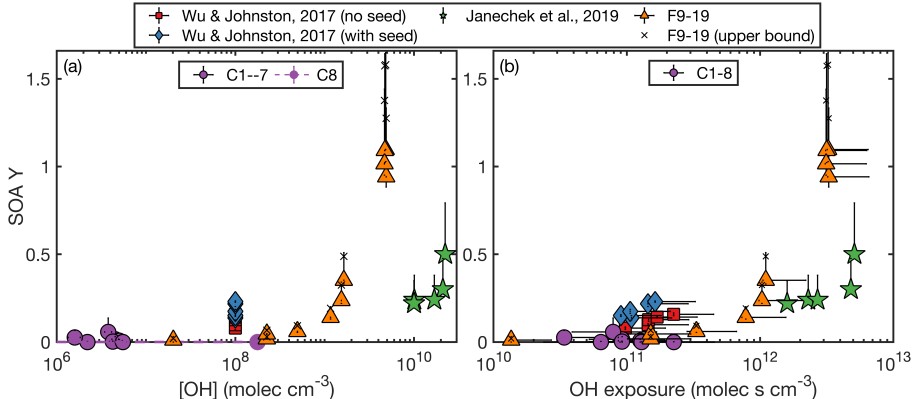

**Figure 1.** Measured SOA yield as a function of the (a) OH concentration and (b) OH exposure for the experiments performed here and by Janechek et al. (2019) and Wu and Johnston (2017). In panel (a), C8, which was performed with methyl nitrite, is shown in purple and not outlined in black and the initial and final OH concentrations are connected with a purple dashed line. The vast majority of this experiment was performed under the lower OH concentration. The same OH concentration is used for all experiments from Wu and Johnston (2017).

All chamber experiments (C1–8) give small SOA yields. The only experiment of these with $Y > 3\%$ is C2, which had a large uncertainty due to aerosol sizes outside the measurement range. It seems likely, then, that at OH concentrations $\lesssim 10^7$ molec cm$^{-3}$, and OH exposures $\lesssim 2 \times 10^{11}$ molec s cm$^{-3}$, the SOA yield of D5 oxidation is $< 3\%$ and close to 1% or 0%.

At similar OH concentrations and OH exposures, the chamber and flow tube data agree. While F9–F19 had uniformly higher OH mixing ratios than C1–7, experiment C8 had an [OH] between F9 and F10–19 at its very beginning (though a lower [OH] after $\lesssim 30$). Since F9 and C1–8 have similarly low SOA yields, there appears to be agreement between the SOA yields when viewed as a function of OH concentration. In Fig. 1a, which shows the relationship between the measured OH concentrations and SOA yields for the experiments performed here as well as those from the literature, the chamber and CPOT experiments coincide (shown in purple circles and orange triangles, respectively). The SOA yield increases above ~5% only at [OH]$\gtrsim 5 \times 10^8$ molec cm$^{-3}$.

There is also agreement between the chamber and the CPOT experiments when SOA yield is viewed as a function of OH exposure, as is shown in Fig. 1b. Experiment F9 has the lowest OH exposure of any other experiments performed in this study, and its low SOA yield matches that of C1–8. F10 and F12 have lower OH exposures than C8 and their measured SOA yields

match those of the chamber experiments. Only starting at OH exposures of $\gtrsim 3 \times 10^{11}$ molec s cm$^{-3}$ does the SOA yield begin increasing significantly.

If OH concentration or exposure were the strict determinant of the SOA yield, F16–17 should give the same SOA yields as F18–19. These experiments do have similar SOA yields, and F16–18 are all within error. If there is a difference, it might be attributed to a dependence on the organic aerosol mass concentration at high OH concentrations or exposures. Viewing SOA yield as a function of organic aerosol mass concentration also gives agreement between the chamber and CPOT experiments, as shown in Fig. 2.

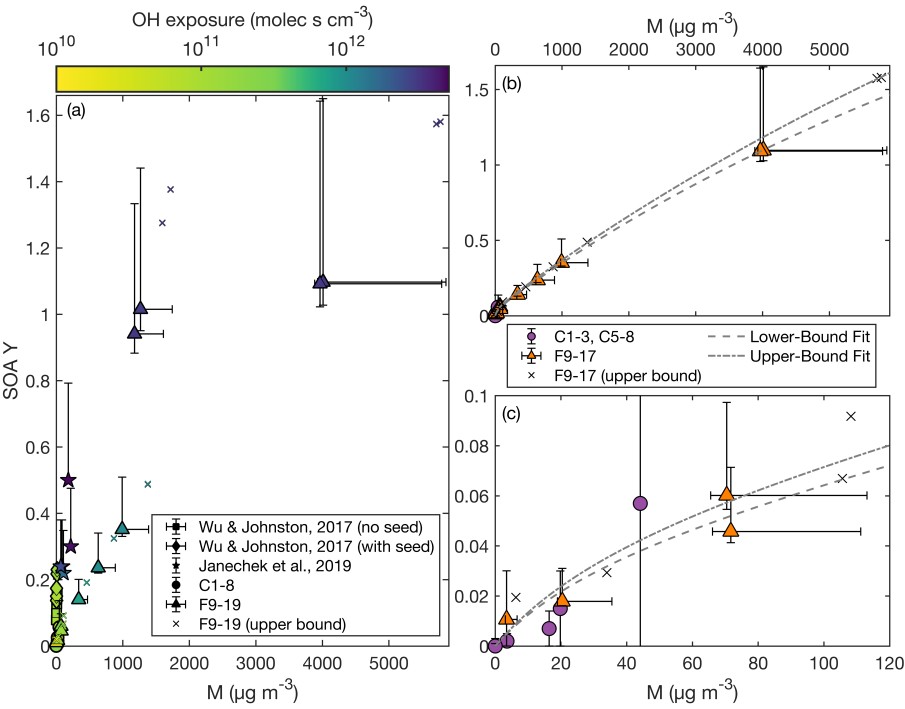

**Figure 2.** SOA yield as a function of organic aerosol mass concentration (M) as (a) compared to that reported by Wu and Johnston (2017) and Janechek et al. (2019). The color axis is the OH exposure. Panel (b) shows the SOA yield for experiments C1–3, C5–8, the non-wall-deposition-corrected F9–17, and the wall-deposition-corrected F9–17. Also included are the fits of a two-product absorption partitioning model to C1–3, C5–8, and the non-wall-deposition-corrected F9–17 (dashed curve) and to C1–3, C5–8, and the wall-deposition-corrected F9–17 (dash and dotted curve). Panel (c) focuses on experiments with M< 120 µg m$^{-3}$.

Assuming an absorptive partitioning model (Odum et al., 1996), a two-product fit was performed for experiments C1–3, C5–8, and F9–17 using Equation 3. This model assumes the same temperature for all experiments, which is why C4 was excluded (conducted at 17.7°C). While the chamber and CPOT experiments also varied in temperature, all the rest were within 5°C of one another. The absorptive partitioning model also assumes that the parent compound is still present; so F18 and F19, where the precursor is completely consumed, were excluded from the fit. A single-product version of Equation 3 did not provide a good fit.

good fit.

The two-product fit that included the non-wall-loss-corrected values for the CPOT experiments gave parameters: $\alpha_1 = 0.044$, $K_{OM,1} = 0.027$ m$^3$ µg$^{-1}$, $\alpha_2 = 5.5$, and $K_{OM,2} = 6.0 \times 10^{-5}$ m$^3$ µg$^{-1}$. For the upper bound, which is a fit with the wall-loss-corrected values for experiments F9–17, $\alpha_1 = 0.056$, $K_{OM,1} = 0.022$ m$^3$ µg$^{-1}$, $\alpha_2 = 7.7$, and $K_{OM,2} = 4.3 \times 10^{-5}$ m$^3$ µg$^{-1}$. Both of these fits are shown in Fig. 2; panel b shows the full range and panel c zooms in on the region with $M < 120$ µg m$^{-3}$.

Note that, as shown in Fig. 2a, the experiments with larger $M$ similarly had larger OH exposures (and OH concentrations), since these were the experiments with higher SOA yields. Also, when all the D5 is consumed, the absorptive partitioning model no longer applies; in Fig. 2a, experiment F15 (triangle with M=991 µg m$^{-3}$) has a similar $M$ as F18 and F19 (triangles with $M$ of 1267 and 1175 µg m$^{-3}$, respectively), but they have very different SOA yields.

    The major difference in F16–17 and F18–19 is the percent of D5 that reacted by the end of the experiment: 97% for F16, 235    98% for F17, and 100% for F18–19. Figure 3 shows the fraction reacted compared to the SOA yield for experiments performed in this study and those in the literature. The color axis in Fig. 3 shows the OH exposure corresponding to the fraction of D5 reacted. This fit could indicate that there are later generation oxidation products that form large amounts of aerosol and that the gas-phase reaction rate to form the low-volatility later-generation oxidation product is slower than the gas-phase reaction rate to form the first-generation product (Kroll and Seinfeld, 2008).

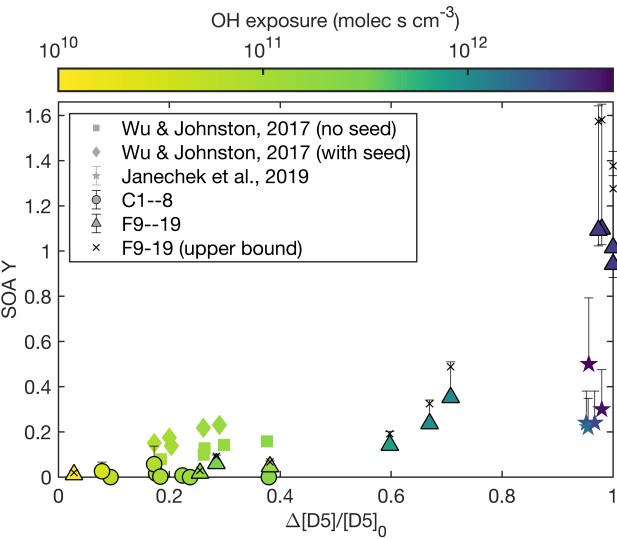

**Figure 3.** Measured SOA yield as a function of the fraction of D5 reacted at the end of the experiment. The color of each point indicates the OH exposure for the experiment. Experiments performed here are circles and triangles and are outlined in black, those by Janechek et al. (2019) are stars, and those by Wu and Johnston (2017) are squares and diamonds. The wall-deposition-corrected data for F9–19 are shown as black Xs.

We do not expect that either relative humidity or temperature affect the SOA yield sufficiently that these would account for the vastly different measured SOA yields under different OH concentration or exposures. Experiments C1–8 were performed at relative humidity (RH) levels between 2 and 6%, experiments F9–19 were between 0 and 30% RH, those by Wu and Johnston

(2017) were performed at 27°C and a RH of 8–10%, and the experiments from Janechek et al. (2019) were run at 24°C and an RH of 25% or 45%. At similar values of relative humidity but different OH concentrations (e.g., F9–12, which all have RH ≤ 6%), the OH concentration or exposure matters for determining the SOA yield. For C3 and C4, the lowest and highest temperatures studied here (17.7 and 27.6°C, respectively), the measured SOA Y varies by < 3%, which is within the uncertainty.

The $NO_x$ concentrations also do not seem to affect the SOA yield for OH concentrations $\sim 5 \times 10^6$ molec cm$^{-3}$, as discussed in Appendix A. While the D5 oxidation chemistry may depend on the NO mixing ratio (but not on the $NO_2$ mixing ratio), this has no effect on the measured SOA yield for the chamber experiments.

## 3.2 Agreement with literature

Also plotted in Figs. 1–3 are the experiments performed by Wu and Johnston (2017) and Janechek et al. (2019). The results from Wu and Johnston (2017), which were used by McDonald et al. (2018) for evaluating the contributions of D5 to aerosol levels in the Los Angeles Basin, were performed in a 50 L PFA photooxidation chamber with reported OH concentrations of $\sim 10^8$ molec cm$^3$ (the error of which "was difficult to assess") at 27°C and a RH of 8–10%. Their data are neither vapor- nor wall-deposition corrected. For similar initial D5 concentrations (and, hence, for similar organic aerosol concentrations), the measured SOA yields were uniformly higher in experiments that were initiated with ammonium sulfate seed than those that were not (see Fig. 1). We, therefore, show the seeded and unseeded experiments in Figs. 1–3 as diamonds and squares, respectively. In Fig. 1, these points are, respectively, blue and red.

OH concentration in the experiments reported by Wu and Johnston (2017) were calculated by replacing the precursor with $SO_2$, measuring the formation of aerosol, and assuming that all the $SO_2$ reacted with OH to form $H_2SO_4$ and all the sulfuric acid formed aerosol with minimal wall loss (Hall et al., 2013). Because of the uncertainties present for each step of this measurement, it seems reasonable that this [OH] estimate could be too low by at least a factor of 2. If the experiments from Wu and Johnston (2017) had OH concentrations more than twice as large, points in Fig. 1a would roughly agree.

Other instrumental and analysis uncertainties might close the gap between the OH concentrations measured by Wu and Johnston (2017) and the OH concentrations found in the experiments performed here. For example, the CPOT experiments and the Wu and Johnston (2017) experiments calculate the total OH exposure experienced in the flow reactor and then find the OH concentration by taking the ratio of this exposure and the residence time. Since the reactor used by Wu and Johnston (2017) is a rectangular bag, regions will exist with differing OH concentrations. If this reactor has slightly higher concentrations in some points or its residence time is overestimated or if the residence time for CPOT is a slight underestimate (we calculated an uncertainty of ~2%), this could account for the remaining disagreement between the data from the two experimental setups.

We calculated the OH exposure in the experiments by Wu and Johnston (2017) in the same manner as for experiments C1–8, and there appears to be fairly good agreement between the SOA yields measured as a function of OH exposure (see Fig. 1b).

Differences in the analysis could change the relevant SOA yields calculated, which could cause better agreement between experiments sets when $Y$ is viewed as a function of both OH concentration and exposure. For F10–19, we measured both the initial and the final D5 concentration and for C1–8 we continuously measured the concentration. Wu and Johnston (2017) measured the initial concentration and calculated the SOA yield by using the [OH] to estimate the amount of reacted D5. If

Wu and Johnston (2017) underestimated the [OH], they might have correspondingly overestimated Y because they would have assumed less D5 reacted than in actuality. To achieve agreement to experiments performed here, then, the [OH] concentration could be different by less than a factor of 2 due to these confounding variables.

We also assumed that the density of the SOA formed was 1.52 g cm$^{-3}$ and Wu and Johnston (2017) collected the aerosol onto filters and directly measured the mass formed. Much secondary organic aerosol has a density of 1.4–1.6 g cm$^{-3}$ (Kostenidou et al., 2007), but deviations from this range could account for some of the discrepancy between the sets of experiments. Additionally, since the CPOT experiments were unseeded, seeded experiments increased the measured Y, which could also have led to better agreement.

Data from Janechek et al. (2019) show the opposite disagreement: OH concentrations and exposures are a factor of ~2 too large to perfectly match the results presented here. Janechek et al. (2019) performed their experiments in a 13.3 L potential aerosol mass oxidation flow reactor (PAM OFR) with OH concentrations on the order of $10^{10}$ molec cm$^{-3}$. They reported the total OH exposure and calculated it similarly to the method used for the CPOT (using Equation 2). We convert this to the [OH] plotted by dividing this OH exposure by the residence time (calculated from the size of the reactor and the reported flow rate)

and assuming that OH concentrations throughout the reactor are approximately constant. Just as with the CPOT experiments, their experiments are unseeded, and they measure the initial and final D5 concentration directly. They used an SOA density of 0.959 g cm$^{-3}$ to calculate Y and the positive error bars shown are an adjustment of their SOA yields to the 1.52 g cm$^{-3}$ used in the experiments performed here.

While they corrected for particle loss downstream of their reactors, they did not account for those particles lost within their

reactor; this could have led to an underestimate of their SOA yields. While the methods to calculate OH concentration and exposure were very similar, the CPOT and the PAM OFR are nevertheless different and, therefore, OH concentrations could vary locally in dissimilar ways between the reactors. A factor of 2 disagreement could be within the uncertainty. A comparison between predicted and estimated OH exposures for the PAM OFR indicates agreement only within a factor of 3 (Li et al., 2015; Janechek et al., 2019), so a factor of 2 disagreement in OH concentrations or exposures would seem to be with the uncertainties

for the CPOT and the PAM OFR.

## 4   Conclusions

The atmospheric aerosol formation potential of D5 was investigated under a range of OH concentrations and exposures. While secondary organic aerosol (SOA) yields can reach 110% (158% at the upper limit) at OH mixing ratios of $\sim 5 \times 10^{9}$ molec cm$^{-3}$ and OH exposures of $\sim 3 \times 10^{12}$ molec s cm$^{-3}$, at lower OH concentrations and exposures ($\lesssim 10^{7}$ molec cm$^{-3}$ and $\lesssim 2 \times 10^{11}$

molec s cm$^{-3}$, respectively), SOA yields do not exceed 6% and are likely ~1%.

Between the experiments performed here and those in the literature, the SOA yields vary widely; but, these SOA yields are correlated to a similarly large range of OH concentrations and OH exposures. Environmentally relevant OH concentrations are on the order of $10^{6}$ molec cm$^{-3}$; since D5 is primarily lost to OH and has a half life of 3.5–7 days, OH exposures on the order

of $10^{12}$ molec s cm$^{-3}$ are also relevant. When viewed as a function of OH concentration or exposure, results here generally agree with those from the literature.

Due to experimental limitations, in particular an inability to perform experiments for multiple days without diluting the sample and otherwise changing the conditions, the OH concentration and exposure are often correlated, as was the case for these experiments. The correlation, however, will be different in the atmosphere than in the lab. When extrapolating these laboratory results or comparing to other studies, atmospheric modelers and experimentalists should be careful about understanding the relevant OH concentrations and exposures because the two variables may have different effects on the chemistry of the system and, correspondingly, the SOA yield. There may also be other variables, not investigated in this work, that affect the chemistry and SOA formation of D5 oxidation.

Regardless of the true, ambiently relevant SOA yield of D5, silicon has still been observed in ambient aerosol and its concentration is likely somewhat population (and not vehicle) dependent (Bzdek et al., 2014; Pennington et al., 2012). If a lower SOA yield than previously thought is atmospherically appropriate, it could be possible that the silicon present is from D5 or other volatile methyl siloxanes, just in lower concentrations than expected, since D5 is so abundant. Another possibility is that silicon in the aerosol-phase comes from polydimethylsiloxanes (Weschler, 1988).

Since the aerosol formed from volatile chemical products (VCPs) may dominate the high concentrations of particulate matter found in urban areas (McDonald et al., 2018), understanding those VCPs that have high aerosol-formation potential and those which do not is important for formulating policy to reduce human exposure to organic aerosol.

## Appendix A: NO$_x$-Dependence of SOA Yield

For the chamber experiments, which had atmospherically relevant OH concentrations, the SOA yield does not change depending on the NO$_x$ concentration: experiments with no NO$_x$ present are on both the lower and higher end of the SOA yields for the chamber experiments. Those with a continuous injection of NO throughout the experiment, which ensured that the NO/HO$_2$ ratio remained high even as the NO reacted, had SOA yields similar to both the no NO$_x$ and the initial NO experiments. This indicates that different NO mixing ratios did not have an effect on the measured SOA yield.

This does not imply that the chemistry is independent of NO concentration. Indeed, the concentrations of gas-phase fragments detected by the CIMS at m/z 139, 169, 243, and 317, which likely correspond to oxygenated fragments of D5, depend on the NO concentration but not the NO$_2$ concentration. Figure A1 shows the signal for these fragments normalized to the reagent ion as a function of the NO concentration at any time. Note that, since some of the methyl nitrite is detected as NO, data from C8 were not included. Figure A2 shows the NO and NO$_2$ concentrations in each experiment as a function of time.

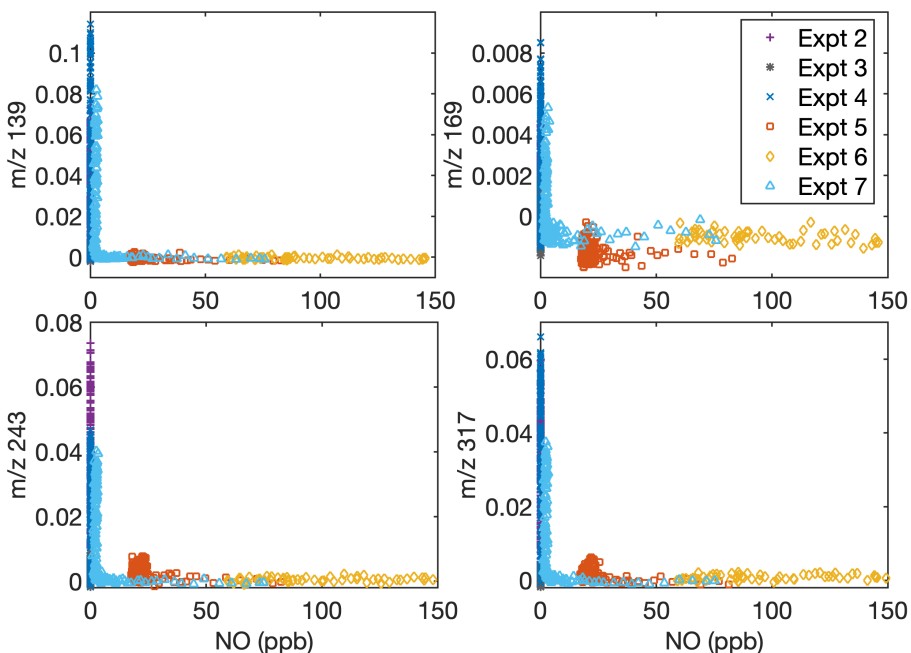

**Figure A1.** Dependence of gas-phase D5 oxidation products on the NO concentration in the chamber indicates that oxidation chemistry changes depending on NO concentrations. Signals normalized to the reagent concentrations with (a) m/z=139, (b) m/z=169, (c) m/z=243, and (d) m/z=317 are shown as a function of NO concentration. Experiment C6 has [NO] extending to >450 ppb, but since the normalized signal remains close to 0, data above [NO]=150 ppb are cut off for clarity. Because of the inaccuracy of NO measurements during oxidation when methyl nitrite is present, C8 is not included.

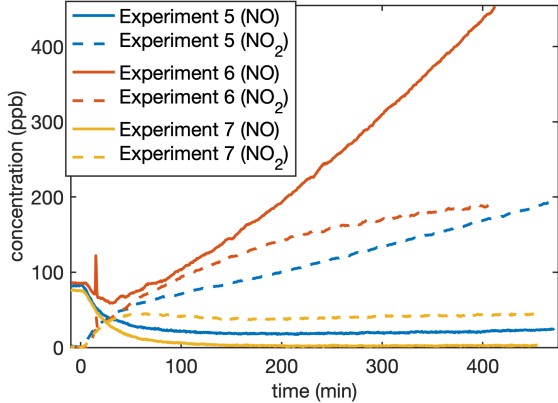

**Figure A2.** For the experiments that included $NO_x$, the NO and $NO_2$ concentrations as a function of the time since the onset of oxidation. C8 is not included, since methyl nitrite was present. The measurement uncertainty is ~5 ppb, but any organonitrates would also be measured as $NO_2$.

Fu et al. (2020) found that the gas-phase rearrangement of methylsiloxanes is dependent on the NO/HO$_2$ ratio. A comparison of Figs. A1 and A3 shows that the concentration of some gas-phase fragments is dependent on the NO mixing ratio but not on the NO$_2$ mixing ratio. This is consistent with gas-phase products depending on the NO/HO$_2$ ratio. Note that at all NO/HO$_2$ ratios investigated, aerosol formation is still minimal when [OH] is small.

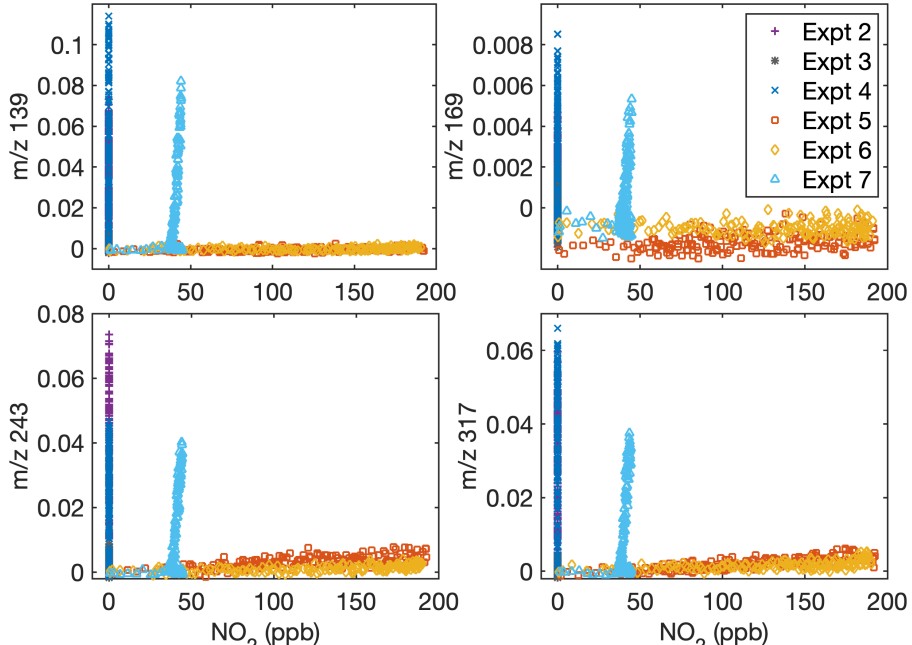

**Figure A3.** Dependence of gas-phase D5 oxidation products on the NO$_2$ concentration in the chamber indicates that oxidation chemistry does not depend on NO$_2$ (but does depend on NO, see Fig. A1). Signals normalized to the reagent concentrations with (a) m/z=139, (b) m/z=169, (c) m/z=243, and (d) m/z=317 are shown as a function of NO$_2$ concentration. Because of the inaccuracy of NO$_2$ measurements during oxidation when methyl nitrite is present, C8 is not included.

*Data availability.* Chamber data available upon request and through the Index of Chamber Atmospheric Research in the United States (ICARUS), experiment sets 220 and 221, https://icarus.ucdavis.edu/experimentset/220 and https://icarus.ucdavis.edu/experimentset/221

*Author contributions.* SMC designed the experiments, carried out the data collection and analysis, and wrote the manuscript. YH assisted with the CPOT experiments. RSB participated in discussions about the study. QL measured the aerosol density. DRC secured funding for the project. JHS supervised the work. All authors reviewed and edited the manuscript.

*Competing interests.* The authors declare that they have no conflict of interest.

*Acknowledgements.* The authors would like to thank Nathan Dalleska for his assistance with the GC-FID; John Crounse for his general help and for synthesis of $CF_3O^-$ for the CIMS; Paul Wennberg for the use of his FT-IR and for his insight during discussions of the system; Lu Xu and Benjamin Schulze for useful input; and Mitchell Alton and Eleanor Browne for helpful discussions. The project was funded by the California Air Resources Board (Contract #18RD009). SMC and RSB were funded by the National Science Foundation Graduate Research Fellowship program (#1745301).

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
