# Peer review of "Supporting Information: Secondary Organic Aerosol Formation from the Oxidation of Decamethylcyclopentasiloxane at Atmospherically Relevant OH Concentrations"

_Atmospheric Chemistry and Physics, 2021_

## Author Comment (AC1)

**Response to Reviews: Secondary Organic Aerosol Formation from the Oxidation of Decamethylcyclopentasiloxane at Atmospherically Relevant OH Concentrations**

Thank you for all of these comments. We have addressed each individually in blue text. Additionally, after submitting this manuscript, we were able to better estimate the density of the secondary organic aerosol (previously, we assume 1.1, now we assume  $1.52 \text{ g cm}^{-3}$ ).

**Reviewer 1**

- 5 This manuscript describes new secondary organic aerosol (SOA) yields from decamethylcyclopentasiloxane (D5) oxidation by OH at a range of OH exposures/concentrations and compares the new results to previous publications. The authors find that the SOA yield varies significantly based on the experimental conditions and argue that differences in OH concentration is the cause of this variation. They stress the importance of understanding the effects of OH concentration versus OH exposure in conducting SOA yield experiments. D5 is a volatile chemical product (VCP), a class of compounds that is of emerging
- 10 interest and importance for air quality. The yields reported here at lower OH concentrations are much smaller than previously published yields and thus previous estimates of SOA from D5 in urban areas may be overestimated. I believe these results merit publication in this journal; however, I think revisions and consideration of additional points are necessary before I can recommend publication.

**Main Comments**

- 15 For me, the main take-home of this work is that the SOA yield for D5 + OH may be substantially lower than previous reports. I think this is an important point and it signifies D5 is a chemical system requiring further investigation. Indeed, there is emerging evidence suggesting that the chemistry of siloxanes differs from more traditional VOCs (e.g., Ren and da Silva, 2020; Fu et al., 2020). As to the cause of the SOA yield varying based on OH exposure/concentration, I think better support as to why/how it varies with OH concentration is necessary if the authors wish to make this a main point of the paper. Personally, I think the
- 20 extended discussion on the role of OH concentration versus exposure detracts from the take-home point of the paper. I think the new yield measurements merit publication even without a detailed investigation of why the yield differs between experiments.

In my opinion, either the focus on the role of OH concentration versus exposure should be deemphasized or a more complete consideration of the chemistry should be included.

We decided to focus on the SOA yields and deemphasize the OH concentration vs. exposure discussion. In the conclusion, we have the following paragraph: "Due to experimental limitations, in particular an inability to perform experiments for mul-25 tiple days without diluting the sample and otherwise changing the conditions, the OH concentration and exposure are often correlated, as was the case for these experiments. The correlation, however, will be different in the atmosphere than in the lab. When extrapolating these laboratory results, atmospheric modelers should be careful about understanding the relevant OH concentrations and exposures because the two variables may have different effects on the chemistry of the system and, 30 correspondingly, the SOA yield."

50

I agree with the point that it is important to understand how the experimental conditions impact SOA yields so that yields can be appropriately extrapolated to the ambient atmosphere. However, I think that the conclusions regarding the experimental issues of high OH versus concentration (lines 266-269) are oversimplified. The current presentation points to high OH concentration as a fatal flaw for this chemical system. However, the OH exposure versus concentration issue is a concern about

- 35 the relative role of processes that scale with OH and are atmospherically relevant versus those that do not scale with OH (e.g., peroxy radical (RO2) isomerization) and/or that might scale with OH but are not typically important for most systems (e.g. RO2 + OH). Consideration and analysis of the chemistry within the experiment are important for determining if high OH experiments will always be difficult for this chemical system (for instance if photolysis or isomerization are important) or if high OH experiments are possible with careful planning (e.g. by running under conditions to limit RO2 + OH; for instance see
- Peng and Jimenez (2020) and references therein). This distinction between high OH experiments being possible but requiring 40 careful experimental consideration versus difficult and unlikely to provide useful information is an important consideration for how the community plans, performs, interprets, and extrapolates chamber and flow reactor experiments. I ask the authors to consider these points when making a recommendation about future experimental conditions.

Yes, we did not mean to imply that this was a "fatal flaw" for this chemical system, only that differentiating between OH concentration and OH exposure in our lab's current set-up is difficult. We've removed this section since we removed most of 45 the OH concentration vs. exposure discussion in the rest of the paper.

In my opinion, the thinking of how OH exposure versus concentration affects this chemical system is poorly articulated. The reasoning outlined in lines 165-167 is confusing to me, at least in part because it seems like first-, rather than second-, generation product is a more appropriate term to use. Please clarify the mechanisms that may be impacting this system. To do this, I believe that further information on the radical chemistry in the chamber and CPOT experiments. For instance, one possibility at high OH is that RO2 +OH becomes important. Does the estimated HO2/OH ratio for the experiments support this idea? Would it be possible to adjust the HO2/OH ratio in the experiments to avoid this condition while still maintaining high OH concentrations? Overall, information on how the RO2 lifetime and fate (isomerization or reaction with NO, HO2, RO2,

OH) varies across the experimental conditions is necessary for the reader to judge if high OH is the fatal flaw it is made out to

55 be. I recognize that investigating this chemistry for D5 is difficult since little is known about the gas-phase chemistry of D5, however, educated guesses are possible and necessary for an exposure versus concentration argument.

We've removed the OH exposure vs. concentration discussion from the manuscript, except for as a side note in the conclusions.

The idea that concentration and exposure are not necessarily interchangeable is well-known from the heterogeneous chem-60 istry literature (e.g., Liu et al., 2011; Renbaum and Smith, 2011; McNeill et al., 2008). For SOA, Lambe et al., (2015) found only small differences between chambers and flow reactors for many systems. Additionally, Peng and Jimenez (2020 and references therein) have investigated this using models and provided recommendations for operation. At least the work comparing SOA between chambers and flow reactors should be discussed.

Thank you for your recommendation to include these relevant papers. We have summarized their work and added the following description to the introduction: 'It is well established that OH concentration and exposure are not necessarily interchangeable (Renbaum and Smith, 2011; Liu et al., 2011; McNeill et al., 2008). Additionally, there is precedent for studying an overlapping range of OH exposure using both environmental chambers and flow reactors. For example, Lambe et al. (2015) showed that oxidation experiments over a range of OH exposures can be comparable between both types of reactors."

I am not convinced that later generation products can be disregarded (lines 235-236). The contribution of later-generation products to the SOA yield is discarded based on a lack of correlation between the yield and the OH exposure normalized to 70 D5 reacted and the finding that for experiments 16-17 where all the D5 does not react has a higher yield than experiments 18-19 where all the D5 does react. However, experiments 16 & 17 have a higher absolute concentration of D5 compared to experiments 18 & 19. Could the results be influenced by RO2 + RO2 reactions leading to lower volatility products and hence more aerosol in 16 & 17 compared to 18 & 19? Although RO2 + RO2 is typically slow, it can be fast for some RO2. 75 Additionally it has been shown that dimers and products containing more than 5 Si are important in D5 generated aerosol (Wu and Johnston, 2016, 2017). While it is unclear if the dimers are formed via gas- or condensed-phase chemistry, those results do suggest that there may be a D5 concentration dependence. Overall, I think a more detailed characterization of the RO2 chemistry and D5 concentration dependence is necessary before higher generation oxidation products are deemed to not matter (lines 235-236). While D5 RO2 + RO2 chemistry generating dimers may be unlikely to occur in the ambient atmosphere (thus 80 reinforcing the point that there needs to be careful consideration of how experimental conditions relate to the atmosphere), this is different from the OH exposures versus concentration argument.

We've removed discussion of which generation products are or are not important. It does not appear that D5 concentration alone is responsible for differing SOA yields, since varying the initial D5 concentration did not appear to affect the SOA yield when other conditions remained similar (e.g., compare Y from C3 and C1).

Lines 114-120: I think the discussion on 5% of the oxidation products being lost to the walls is somewhat misleading. While I agree that first-generation oxidation products such as the ester will have minimal wall-loss, they will also contribute minimally to aerosol if absorptive partitioning dominates. Later generation products may have higher wall-loss.

The goal of this comment was to point out, even with some degree of wall loss, the SOA yields would still be small. We've 90 changed part of this section to:

"Though other products may have higher wall-loss rates, if 5% of the oxidation products were lost to the chamber walls in C1–8, the SOA yields would still be within the reported uncertainty and smaller than expected previously."

**Technical**

Figures 1-3: Colors and shapes are hard to distinguish, particularly in the legend and for the red squares and blue circles in Fig. 95 1a. Perhaps removing the black outline (or making it thinner) and/or making the points bigger would help.

The problem in Fig. 1a is that all the experiments from Wu and Johnston, 2017 have the same OH concentration. Regardless of the size of these points, they will overlap and be difficult to see. Without the black outline, all of these points run together, making it impossible to tell that there are multiple points present. We have made the shapes bigger so that the rest of the figure is clearer and added the line "The same OH concentration is used for all experiments from Wu and Johnston (2017)." We also changed the cap size of the error bars, so that they are less distracting but still informative.

The point of the plots is to see the general trend and not the individual points, which we think is still possible with the overlap currently present.

Since 1b uses the same color and shapes, we think it's important to keep the red squares and blue diamonds on Fig. 1a for comparison.

105 Please include the RH for the experiments in Table 1.

Relative humidity is now in Table 1.

100

**Reviewer 2**

This paper describes chamber and flow-reactor measurements of secondary organic aerosol (SOA) yields from the OH oxidation of decamethylcyclopentasiloxane (D5), an important organic compound in indoor environments. There are two major

- 110 results from this work: (1) chamber yields are much lower than previous measurements, whereas flow-reactor yields are quite high, and (2) this difference is attributed to the use of atmospherically unrepresentative OH levels in the flow-reactor experiments. Both of these are important topics, and certainly will be of interest to the readership of ACP. However the manuscript focuses mostly on the 2nd result; I think the paper would be stronger if it spent more time discussing the 1st one (the yields themselves) as well. This could include providing SOA yield parameterizations for use in modeling, and/or describing impli-
- 115 cations for indoor/urban air quality. In addition, I had a number of questions and comments about the 2nd conclusion (OH concentration and exposure); these are described below.

The use of the CPOT was primarily to explore why chamber SOA yield results were so small and result (2) emerged from a desire to understand result (1). In response to your (and the other reviewers') comments, we have removed the discussion of result (2). We also added fits to a two-product absorptive partitioning model for use in modeling, to make result (1) more helpful for modelers as suggested.

1. Much of the manuscript is focused on examining the role of OH concentrations vs. OH exposure; however this discussion (or the possible underlying mechanism) was not always clear:

166-169: I had a hard time following this explanation. This is in part because the language is very general, mentioning "intermediates" and "fragments" but not providing concrete examples. Are these intermediates/products radicals or molecules? And why is the focus here on only second-generation (but not first-generation) products? Some more discussion of specific pathways (e.g., oxidation of products, RO2 self-reactions, oligomerization reactions, etc.) would be helpful.

**We've removed this discussion.**

Similarly, a major conclusion of the work (line 265) is that "It is the OH concentration, and not the OH exposure, that affects the SOA yield"; but little explanation for why this might be the case. Can the authors suggest some potential mechanisms? (The CIMS data might be of use here.)

**We've removed this conclusion.**

135

120

125

130

My initial interpretation of the data (especially Fig 2) was that the later-generation condensable products are what lead to SOA formation; since these may not be formed until most D5 reacted, yields will be higher when deltaD5 / initialD5 approaches 1. The implication of this would be that yields are highly dependent on extent of reaction. Lines 226-236 argues against this interpretation, but I don't follow the argument. Differences in fractions of D5 reacted between experiments 16-17 and 18-19 are described, but these differences are quite small (97-98% vs 100%). A much bigger difference is the absolute amount of D5 reacted ( 240 ppb vs ~80 ppb) – this could lead to large differences in aerosol

loadings, which in turn could affect yields due to differences in semivolatile partitioning. The oxidation chemistry might be different as well. Given such large differences, it doesn't seem straightforward to make any conclusions about SOAformation mechanisms just from comparisons of yields or fraction of D5 reacted.

We no longer make conclusions about the SOA-formation mechanism.

140

150

155

160

165

(Also, the statement "the fraction of D5 reacted correlates with the [OH]" (line 235) seems self-evident, given the role of OH concentration in reacting away D5, as shown in Equation 1.)

145 We have changed the sentence to "The color axis in Fig. 3 shows the OH exposure corresponding to the fraction of D5 reacted."

There's a good deal of literature on the possible nonlinearities associated with OH exposure (that [OH] cannot always be ramped up to accurately simulate long atmospheric timescales), (examples include Renbaum and Smith, Atmos. Chem. Phys. 2011, 11, 6881–6893, Liu et al. 2011, PCCP, 13, 8993-9007, and Palm et al. 2016, Atmos. Chem. Phys. 16, 2943–2970), but these aren't discussed in this paper. It's probably worth discussing these treatment in the context of the present results.

Thank you for the recommendations. We have now included a summary of the literature and have cited some of these papers in the introduction: "It is well established that OH concentration and exposure are not necessarily interchangeable (Renbaum and Smith, 2011; Liu et al., 2011; McNeill et al., 2008). Additionally, there is precedent for studying an overlapping range of OH exposure using both environmental chambers and flow reactors. For example, Lambe et al. (2015) showed that oxidation experiments over a range of OH exposures can be comparable between both types of reactors."

The authors argue for a "the necessity of OH concentrations similar to the ambient environment when extrapolating SOA yield data to the outdoor atmosphere." What are the implications of this work for other laboratory studies? The last several years has seen a huge increase in the use of oxidation flow reactors for studying SOA chemistry (e.g., Chem. Soc. Rev., 2020, 49, 2570-2616). Is the argument then that these results are flawed, and should not be used in models?

We do not mean to imply that all results from oxidation flow reactors are flawed. We removed this phrase along with most of the discussion about OH concentration vs. exposure. We've replaced it with softer language about modelers carefully choosing how to extrapolate data: "Due to experimental limitations, in particular an inability to perform experiments for multiple days without diluting the sample and otherwise changing the conditions, the OH concentration and exposure are often correlated, as was the case for these experiments. The correlation, however, will be different in the atmosphere than in the lab. When extrapolating these laboratory results, atmospheric modelers should be careful about understanding the relevant OH concentrations and exposures because the two variables may have different effects on the chemistry of the system and, correspondingly, the SOA yield."

170 2. A central point of the paper is the relative importance of OH exposure and OH concentration; I have a number of comments related to the calculation of these quantities:

144-146: I'm unclear on how OH exposure was estimated using SO2. At first I assumed SO2 was added to the reactor (as is sometimes done in OFR experiments), but instead from the text it appears that these SO2 levels were taken from a different set of experiments from a different laboratory (Janechek et al. 2019); this should be described in greater detail. If that is indeed the case, I think it's unlikely that the experiments were "identical" – they presumably used a different reactor (with different flows, concentrations, SA/V ratios for wall loss, etc.). And if SO2 is high enough (relative to the D5) it can affect (decrease) the OH exposure. Therefore such an estimate may well lead to large errors in estimated OH exposure, and probably shouldn't be used in a quantitative way (at least without large error bars).

- To clarify, the experiments were performed in CPOT using D5 and not  $SO_2$ . Immediately after each experiment set, all flows were kept the same, except that D5 was not injected and instead  $SO_2$  was injected into the same reactor (CPOT). Results from the experiments with  $SO_2$  in CPOT were used to determine the OH exposure from the "identical" experiments with D5 in CPOT, where "identical" means that only D5 and  $SO_2$  changed (and they were performed, in many cases, on the same day). The citation given (to Janechek et al. 2019) was to specify that the method was the same as used in that paper and not to indicate that the  $SO_2$  experiments were conducted in the reactor used in that paper.
- 185 We changed the text so it now reads: "...for a CPOT setup with SO2 instead of D5 and otherwise identical flows and conditions, as in the method described by Janechek et al. (2019)."

Additionally, the concentrations used in these experiments are low enough not to affect the OH exposure.

149-150: I don't think these are possible explanations, since Equations 1 and 2 should hold regardless of the sources and sinks of OH.

190 The reviewer is correct that Equations 1 and 2 hold in the calculation of OH exposure. The key point is that Equations 1 and 2 calculate the OH exposures of the single compound. If there exists any additional OH reactivity in the system, or if the OH reactivities of D5 (including from its products and possible interfacial OH reaction on SOA) and SO2 are not similar, the OH concentration will be different. Though we keep the OH reactivity of D5 and SO2 similar for the OH exposure experiments, the later generations of D5 products and SOA surface can be extra sinks of OH, thus lowering OH concentration (and the OH exposure of D5). We have rewritten the sentence as: "This effect may be because of a lower OH concentration in the D5 system due to the reaction of OH with the later D5 oxidation products or the aerosol surfaces."

For calculating OH exposure from Equations 1 and 2, were these single point calculations (estimating exposure from one time point, t), or were they fit to a curve? This latter approach (which is possible at least for the chamber experiments) would give a more precise value.

For the CPOT experiments, OH exposure was a single-point calculation, because we measure concentrations only at the inlet and outlet. For the chamber experiments, the OH concentration was calculated with all points on the curve. The OH exposure was calculated only with the single beginning and end points. We chose this approach because there was noise in the [D5] data, so the initial D5 concentration (which is the average of many measurements, prior to the beginning of oxidation) has less uncertainty than any of the individual measurements.

205

200

175

While a fit to a first-order exponential (as with the OH concentration measurements for the chamber experiments) benefits from having as many points as possible, these fits were less successful at capturing the very end of the chamber experiments, especially for experiments 5–8. This could be because a nearly constant OH concentration decreased slightly by the end of the experiment.

For experiment 8, where the OH concentration changes dramatically throughout the experiment, using a fitted endpoint instead of the actual endpoint doesn't make sense.

215

225

Other fits, which could potentially smooth the noise of the endpoint, but would be based on an arbitrary functional form, didn't appear necessary because, for experiments 1–7, using the first-order exponential fit with its initial point fixed for the endpoint gives OH exposures within  $2 \times 10^{10}$  molec s cm-3. Since this is well within the reported uncertainty ( $4 \times 10^{10}$ molec s cm-3 in the negative direction and the OH exposure value in the positive direction), especially when considering additional uncertainties in this measurement regarding the uncertainty of  $k_{OH+D5}$  (see below), we've elected to keep the non-fitted endpoint as the second point in the two-point calculation. We believe the non-fitted value is a more accurate representation of the total amount of OH to which the precursor was exposed.

142: the value of k\_OH+D5 used was from one study, but as noted in line 126 other studies have found values than span
a factor of 2 (Atkinson et al: 1.55e-12 cm3 molec-1 s-1, Safron et al 2.6e-12 cm3 molec-1 s-1, etc). This in itself adds a factor of 2 uncertainty to the OH exposure estimate.

Yes, this does make the OH exposure estimate more uncertain. The  $k_{OH+D5}$  from Alton and Browne (2020) has a reported uncertainty of  $1 \times 10^{-13}$  cm3 molec-1 s-1, which is what we use. They compare their results to the others that are off by as much as a factor of 2 (in Fig. 2 of that paper). Though they also don't know the reason for disagreement, they attribute the discrepancies to potential silicon-coatings on reactors or errors in the reference compound reaction rates. So, we find that this is the most reasonable rate to use, since they focused on measuring volatile methyl siloxanes and ensuring their reactors were free of contaminants. Furthermore, the uncertainties we use for the OH exposure that are calculated with  $k_{OH+D5}$  are already high:  $4 \times 10^{10}$  molec s cm-3 and the value of OH exposure reported for the negative and positive directions of the chamber experiments, respectively.

For clarity, we added the following sentence in the discussion of OH exposure uncertainty in methods: "This does not include the potential additional uncertainty in the value of  $k_{OH+D5}$  as calculated using similar methods in different laboratories."

Based on the above comments, I believe there's substantial uncertainty in the estimated OH exposures in this study. These should be included as error bars in Fig 1 and 2, and incorporated throughout the text in the discussion of results.

Since Figs. 2 and 3 have OH exposure (previously, it was OH concentration) only in the color bar, there is no way to include error bars without sacrificing clarity. We have large OH exposure error bars in Fig. 1b (which previously was OH exposure/\Delta[D5] and also had error bars), please see the response to the previous question.

29: calculate -> determine?

**240 We've made this change.**

76: why were the concentrations used to calibrate the FID so much higher than those used in the experiment? What sort of error in SOA yields might this lead to?

We have found that the most accurate way to calibrate the GC-FID is to use an FT-IR. While it is not ideal that the detection limit of the FT-IR is higher than the experiments performed, we can measure the true concentration from the FT-IR. This means that we don't have to worry about losses to the injection lines or to the small Teflon bag used for calibration. Since the GC-FID should respond linearly to the concentration, which we also tested, we're not too concerned about calibrating so far from the concentrations used in the experiments.

Table 1: it would be helpful to also give the RH and fraction of D5 reacted for each experiment.

**We've added both RH and fraction reacted at the end of the experiment to Table 1.**

250 121-122: While I agree that wall-loss corrections would not change the conclusions of the paper substantially, a steadystate reactor can still have vapor-phase losses. This can result from extremely low-volatility species, which can sorb to walls essentially irreversibly; also see Krechmer et al. Environ. Sci. Technol. 2020, 54, 12890–12897. (In addition, the text on line 202 implies that vapor wall loss in the flow reactor could indeed lead to a suppression of SOA yields.)

Yes, we agree that a steady-state reactor can have irreversible loss and that the Wu and Johnston (2017) data indicate that some may occur in their continuously run experiments. We've changed "in equilibrium" with "sufficiently high volatility and in equilibrium" and added the following sentences, so that the section now reads:

"The CPOT reactor is operated at steady-state (see Fig. S2) and, therefore, any oxidation products that are sufficiently high volatility and in equilibrium with the bulk flow (i.e., not lost permanently to the quartz walls) do not need a vapor-wall-deposition correction. While very low-volatility compounds may be lost to the reactor's walls, as was seen by Krechmer et al.
(2020) in a continuously run Teflon reactor, we do not expect there to be significant irreversible gas-phase wall loss of siloxanes or their oxidation products in this reactor. Note that Wu and Johnston (2017) did see higher SOA yields in seeded experiments in their steady-state PFA Teflon reactor than in unseeded ones, indicating that there may be some irreversible loss even when run continuously."

204-205: The comparison should probably be in terms of the dependent variable (SOA yield) not the independent one 265 ([OH]).

Since we are comparing across datasets, and we did not perform experiments at the OH concentrations of Janechek et al. (2019), we believe this is a clearer way of describing the figure.

219-221: I'm unclear on how to interpret the "OH exposure divided by reacted D5" metric; this is a nonstandard metric so some more explanation would be useful. The explanation on the following sentence ("the number of OH radicals available per reacted D5 molecule") isn't quite right since that should be unitless, whereas exposure/deltaD5 has units of time

We've removed this metric from the manuscript.

251-260: The case is made that the yield differences are not from differences in [NO] or RH. Might the oxidation conditions (O3 vs H2O2) play a role?

This is a conclusion only for the chamber experiments (C1-8). which had similar H2O2 and O3 concentrations.

We discuss in the methods section (previously line 131) that "The ozone concentration did not affect the SOA yield results: C7 and F9, which were performed at substantially different  $O_3$  concentrations, still gave similar results for the SOA yield (0±0.1% and 0.8±0.8% with an upper wall-deposition-corrected bound of 1.4%, respectively)."

294: I had to look up the "ICARUS" database (no URL was given); it appears not to have been updated since 2019 (with most data being from 2016-2017) and so apparently is no longer active.

280 We added the following information: "Chamber data available upon request and through the Index of Chamber Atmospheric Research in the United States (ICARUS), experiment sets 220 and 221, https://icarus.ucdavis.edu/experimentset/220 and https://icarus.ucdavis.edu/experimentset/221"

The database was down for some time, but everything should now be accessible. To access the database, one may have to create an account (though this is free).

**285 Reviewer 3**

270

This is an important contribution to the literature on the oxidation of volatile methyl siloxanes, to the literature on aerosol formation from volatile chemical products, and on the interpretation of experiments on SOA formation done at high concentration. It should be published in ACP after addressing the following points.

**Major Points**

300

315

- 290 1. The abstract should be revisited, particularly by the senior authors with comprehensive experience across all the aspects of the study (modeling implications, flow tube experiments, chamber experiments). I think the major contributions of the work is that (a) both chamber and flow tube experiments were conducted, (b) a wide range of OH concentrations were looked at experimentally, and (c) these were analyzed in a comprehensive manner with two other datasets from literature. None of these aspects really come out in the abstract clearly.
- 295 We have accordingly edited the abstract.
  - 2. The abstract is confusing chemical transport models and emission inventories: "SOA yields used in emission and particulate matter inventories."

This sentence is no longer part of the abstract.

3. The results start very abruptly and the text at line 156 is not doing anything other than referring the reader to the table. The reader has to do all the work themselves. Could the authors help out and provide the important context?

We have added extensive text to the beginning of the results section.

- 4. Line 166 is important. I quote the manuscript here: "If a chemical process occurs in which the reaction of D5 and OH forms an intermediate or a second-generation product that then either reacts with OH or fragments, then the competition between the two outcomes is moderated by the relative time required for self-reaction or reaction with OH."
- 305 (a) I think this could be written more clearly.

We've removed this discussion.

(b) Line 166 says fragments but 167 self-reaction. Are these same process?

**We've removed this discussion.**

(c) the OFR literature shows that the community is aware of this issue. The concern, however, is usually on the other side
 - that high OH will lead to fragmentation reactions at the gas-particle interface, or that high OH will lead to fragmentation reactions in the gas phase on similar timescales as condensation of low volatility gas phase products ... not that high OH will lead to particle formation while NOT reacting with OH will lead to fragmentation and/or high volatility compounds. This is an interesting route and a valuable contribution to bring it up.

Thank you for bringing this topic up. Given the complexity of the fragmentation reactions, we have decided to remove the discussion on the effect of OH reactivity of later D5 oxidation products.

(to clarify, a and b are reviewer comments where I am looking for a response and/or change to the MS. c is just a reviewer comment)

5. Comment: I think the paper has a critical point on the possible role of fragmentation reactions (or reactions that simply lead to high volatility products) that are zero order in OH and in reaction partners that are correlated with OH. Thus, if first generation oxidation products have such pathways (e.g. rearrangement or autooxidation, for example, leading to high volatility products or to conformations that prevent subsequent formation of low volatility products), and these do not involve OH (but the formation of low volatility products does involve multigenerational oxidation with OH), then there will be a strong OH concentration dependence on yield. For example, if the first generation product of OH attack rearranges to a volatile species with a characteristic time of n seconds, and to form low volatility multi-generational products, OH needs to make a 2nd attack prior to those n seconds, then yields will be dependent on OH and independent of OH exposure. However, there are other mechanisms that could lead to similar dependences, such as concentration effects in the gas-phase, the particle phase, and the gas-particle interface leading to higher order effects not seen at low concentrations.

Reviewer request: acknowledge that there are more mechanism options that could lead to the same functional dependence on concentration of OH.

We have removed the discussion of the mechanism of SOA formation.

6. Regarding fates of the first generation oxidation products that are zero order in OH and in reaction partners that are correlated with OH, but that lead to high volatility products or to conformations that prevent subsequent formation of low volatility products (I believe this is what is required for the strong OH dependence) ... is there evidence, perhaps from the gas-phase mass spec, of such reactions? If so, please report hypothetical reactions and compounds.

We've removed discussion of the chemical mechanism of OH dependence and SOA formation from the manuscript.

7. Comment: The paper implies that reactions (or SOA formation scenarios) can be divided into a class where OH exposure matters, and another where OH concentration is dominant. Not sure this is a useful designation. It is more complicated than that, and it is the fate of the species in the reactor (fixed or flow) that matters, not strictly the OH concentration. I believe the atmospheric chemistry communities that use both types of tools are increasingly aware of the strength, weaknesses, and factors to look out for in data interpretation and extension to models (Lambe et al. 2012, Lambe et al. 2015, Palm et al. 2016, Peng et al. 2016). Reviewer request: reconsider the classification, or provide support for the binary classification.

We've removed this binary classification from the text.

- 8. Abstract line 10, "necessity of OH concentrations similar to the ambient" is probably overstated. There is a literature on the necessary conditions for atmospheric relevance of OFR / PAMS / flow tube type experiments, and this is taking an end run around that literature and oversimplifying the requirements for atmospheric relevance. We've removed this phrase.
  - 9. The introductory material on flow reactors versus chambers is underdeveloped and would benefit from a rewrite and citation to the literature on the topic (e.g., line 40, "researchers use both flow reactors and chambers").

350

330

335

340

We have added to this discussion in the introduction.

**Minor points**

355

360

365

375

1. The paraphrase of Coggon et al. (2018) at line 19 is not quite accurate and should be revisited.

We changed this paraphrase to: "and this dependence is sufficiently reliable that it can be used to tease out personal care product emission patterns (Coggon et al., 2018)."

2. Not sure "Cl is negligible" in the correct interpretation of Alton et al. (2020). The global value was 4.6% for D5, but up to 25-30% for Toronto and Boulder under certain conditions.

This sentence now reads "Given the abundance of D5 in the ambient atmosphere, it is important to understand its fate. The major loss source of D5 is reaction with the hydroxyl radical; losses by reaction with  $NO_3$  and  $O_3$  are negligible (Atkinson, 1991; Alton and Browne, 2020) and global losses by reaction with Cl are less than 5%, though can be higher in polluted areas (Alton and Browne, 2020)."

- 3. As an update to Hobson et al. (1997) on deposition sensitivity, one could refer to the work of Janechek et al. (2017), where dry deposition was quantified and dry deposition modeling parameters were discussed in detail.
- We've kept the reference to Hobson et al. (1997) as it's the citation for the second half of the sentence ("methylsiloxanes do not photolyze in the actinic region"), though we've also verified the lack of photolysis in the CPOT. We've added the Janechek et al. (2017) citation for the negligible wet and dry deposition calculations (in their Section S2).
  - 4. The megacity vs. non-megacity distinction at line 38 is not clear.

The sentence now reads "VCPs are a major (and perhaps majority) source of secondary organic aerosol in some U.S. cities"

5. The foundational work in the UNC chamber on organosilicon oxidation should probably be mentioned in the introduction (Latimer et al. 1998, Chandramouli et al. 2001).

We've added these citations and summary to the introduction.

6. Explain "batch mode" line 45

We've added the following sentence: "Batch mode is where all reactants are added before oxidation and, during each experiment, the evolution of the reactor's contents are tracked in time."

7. Introduction should make it clear that the paper presents new results from both a flow reactor and chamber were used, and give a roadmap to the different sections of the paper.

We added further discussion on flow reactors versus chambers to the introduction. Additionally, it ends with the following roadmap: "We start with a discussion of results from these two reactors, which show agreement when either the OH

- 380 concentrations or exposures overlap. Then, we provide two-product absorptive partitioning parameters and fits for the data collected here. We close by comparing these results to other SOA yield studies in the literature: that of Wu and Johnston (2017) and Janechek et al. (2019)."
  - 8. Line 104 perhaps state (list a paper or report) where a figure can be found with that static mixer shown.

At the end of this paragraph, we added the sentence: "Diagrams of the CPOT and its static mixer are in Huang et al. (2017)."

9. Line 113 – could a time series of a typical chamber experiment be included in SI, showing the steady state stability of the precursor implied at line 113 (negligible wall loss).

We believe the reviewer is asking for a typical CPOT experiment, and so added a figure at the end of the paper that shows the D5 concentration,  $O_3$  concentration, SOA mass concentration, and absolute humidity as measured at the CPOT output.

10. I respect and value the important work on uncertainty quantification, but placing it in the caption of the Table is not ideal. Relocate to methods and/or results and/or SI as appropriate.

We added the information in Table 1's caption to relevant parts of the Methods section.

11. Having explicit labeling of chamber vs. CPOT would be helpful, so that one does not have to know which device is associated with which experiment number.

Experiments conducted in the chamber, run in batch mode, are now labeled C1–8. Those from the CPOT, run at steadystate with a continuous flow are labeled F9–19.

395

**References**

400

Alton, M. W. and Browne, E. C.: Atmospheric Chemistry of Volatile Methyl Siloxanes: Kinetics and Products of Oxidation by OH Radicals

- and Cl Atoms, Environ. Sci. Technol., 54, 5992–5999, https://doi.org/10.1021/acs.est.0c01368, 2020.
- Atkinson, R.: Kinetics of the Gas-Phase Reactions of a Series of Organosilicon Compounds with OH and NO3 Radicals and O3 at 297  $\pm$  2 K. Environ. Sci. Technol., 25, 863–866, https://doi.org/10.1021/es00017a005, 1991.
- Janechek, N. J., Marek, R. F., Bryngelson, N., Singh, A., Bullard, R. L., Brune, W. H., and Stanier, C. O.: Physical Properties of Secondary Photochemical Aerosol from OH Oxidation of a Cyclic Siloxane, Atmos. Chem. Phys., 19, 1649-1664, https://doi.org/10.5194/acp-19-1649-2019, 2019.
- 405
  - Krechmer, J. E., Day, D. A., and Jimenez, J. L.: Always Lost but Never Forgotten: Gas-Phase Wall Losses Are Important in All Teflon Environmental Chambers, Environ, Sci. Technol., 54, 12890–12897, https://doi.org/10.1021/acs.est.0c03381, 2020.
  - Lambe, A. T., Chhabra, P. S., Onasch, T. B., Brune, W. H., Hunter, J. F., Kroll, J. H., Cummings, M. J., Brogan, J. F., Parmar, Y., Worsnop, D. R., Kolb, C. E., and Davidovits, P.: Effect of oxidant concentration, exposure time, and seed particles on secondary organic aerosol
- 410 chemical composition and yield, Atmos. Chem. Phys., 15, 3063–3075, https://doi.org/10.5194/acp-15-3063-2015, 2015.
  - Liu, C.-L., Smith, J. D., Che, D. L., Ahmed, M., Leone, S. R., and Wilson, K. R.: The direct observation of secondary radical chain chemistry in the heterogeneous reaction of chlorine atoms with submicron squalane droplets, Phys. Chem. Chem. Phys., 13, 8993–9008, https://doi.org/10.1039/c1cp20236g, 2011.
- McNeill, V. F., Yatavelli, R. L. N., Thorton, J. A., Stipe, C. B., and Landgrebe, O.: Heterogeneous OH oxidation of palmitic acid in sin-415 gle component and internally mixed aerosol particles; vaporization and the role of particle phase. Atmos. Chem. Phys., 8, 5465–5476. https://doi.org/10.5194/acp-8-5465-2008, 2008.
  - Renbaum, L. H. and Smith, G. D.: Artifacts in measuring aerosol uptake kinetics: the roles of time, concentration and adsorption, Atmos. Chem. Phys., 11, 6881-6893, https://doi.org/10.5194/acp-11-6881-2011, 2011.

Wu, Y. and Johnston, M. V.: Aerosol Formation from OH Oxidation of the Volatile Cyclic Methyl Siloxane (cVMS) Decamethylcyclopentasiloxane, Environ. Sci. Technol., 51, 4445–4451, https://doi.org/10.1021/acs.est.7b00655, 2017.

420

---

## Author Response (AR2)

**Response to Reviews: Secondary Organic Aerosol Formation from the Oxidation of Decamethylcyclopentasiloxane at Atmospherically Relevant OH Concentrations**

Line 44: I don't understand what is meant by "SOA morphology from all D5 oxidation products...". To me this sentence implies that D5 oxidation products were investigated individually. From my understanding of the referenced paper, it was SOA from D5 oxidation that was investigated.

The word "morphology" was removed from this sentence.

Line 201: "Ideally, we seek to report the OH exposure excluding any regeneration." This sentence is confusing since the discussion on regeneration has been removed. It is also unclear to me why regenerated OH should be excluded from the calculation. It would also contribute to oxidation and would thus affect the chemistry leading to SOA. I think this sentence could probably be removed, but if it remains, the reason for excluding regeneration should be clarified.

We removed this sentence.

Figure 2 caption: In the last sentence, panel c (M < 120 ug/m3) should be referenced, not panel b.

This error was fixed.

Figures 2 & 3: Please consider increasing the size of the symbols in the legend to increase readability.

Due to the parameters of making graphics in MATLAB, we elected to leave these legends as is.

Line 393: I like the addition of this paragraph. It explicitly highlights considerations that are important but are often overlooked. I think though that the idea is relevant to both modelers using the data as well as experimentalists designing experiments or comparing to other work. Please consider adding a sentence or a clause about experimentalists.

This sentence now begins: "When extrapolating these laboratory results or comparing to other studies, atmospheric modelers and experimentalists should be careful about understanding the relevant OH concentrations and exposures..."

Figure S1: Please consider reversing the color scale to improve readability by being more consistent with the color scales in the main text (where darker colors were higher values).

We switched the color scale.